# Targeting netrin-1/DCC interaction in diffuse large B-cell and mantle cell lymphomas

Laura Broutier[1], Marion Creveaux[1,†], Jonathan Vial[1,†], Antonin Tortereau[1,2,†], Jean-Guy Delcros[1], Guillaume Chazot[1], Mark J McCarron[3], Sophie Léon[4], Céline Pangault[5,6], Nicolas Gadot[7], Amélie Colombe[4], Marie-Laure Boulland[6], Jonathan Blachier[1], Julien C Marie[3], Alexandra Traverse-Glehen[8], Olivier Donzé[9], Catherine Chassagne-Clément[4], Gilles Salles[10], Karin Tarte[5,6], Patrick Mehlen[1,*] & Marie Castets[1,**]

## Abstract

DCC (*Deleted in Colorectal Carcinoma*) has been demonstrated to constrain tumor progression by inducing apoptosis unless engaged by its ligand netrin-1. This has been shown in breast and colorectal cancers; however, this tumor suppressive function in other cancers is not established. Using a transgenic mouse model, we report here that inhibition of DCC-induced apoptosis is associated with lymphomagenesis. In human diffuse large B-cell lymphoma (DLBCL), an imbalance of the netrin-1/DCC ratio suggests a loss of DCC-induced apoptosis, either via a decrease in DCC expression in germinal center subtype or by up-regulation of netrin-1 in activated B-cell (ABC) one. Such imbalance is also observed in mantle cell lymphoma (MCL). Using a netrin-1 interfering antibody, we demonstrate both *in vitro* and *in vivo* that netrin-1 acts as a survival factor for ABC-DLBCL and MCL tumor cells. Together, these data suggest that interference with the netrin-1/DCC interaction could represent a promising therapeutic strategy in netrin-1-positive DLBCL and MCL.

**Keywords** apoptosis; dependence receptors; non-Hodgkin lymphoma; DCC; netrin-1

**Subject Categories** Cancer; Haematology

## Introduction

Resistance to cell death through silencing of pro-apoptotic signaling cascades or constitutive activation of anti-apoptotic pathways is usually considered as an important step toward tumor progression (Hanahan & Weinberg, 2000). Historically, Bcl-2 was characterized as the first anti-apoptotic protein acting as an oncogene in B-cell lymphoma (Reed, 2008). B-cell lymphoma refer to a heterogeneous subgroup of lymphoid neoplasms, which originate within B cells at different stages of differentiation and are classified into specific entities based on clinical, histological, immunological, and genetic features. Besides Bcl-2, several other factors involved in the regulation of the balance between cell death and cell survival have now been causally linked to B-cell lymphomagenesis (Voutsadakis, 2000; Rummel *et al*, 2004; Muris *et al*, 2006).

DCC (Deleted in Colorectal Carcinoma) is a transmembrane receptor, which actively triggers cell death when disengaged from its ligand netrin-1, thereby functioning as a dependence receptor (DR) (Mehlen *et al*, 1998; Mazelin *et al*, 2004). As such, DCC could act as a conditional tumor suppressor by triggering apoptosis of cells in excess when levels of netrin-1 are limiting. Along this line, it was recently reported that conditional inactivation of DCC expression or invalidation of DCC-induced apoptosis by introduction of a D1290N point mutation is, respectively associated with breast and colorectal tumor progression in two mutant mice models (Castets *et al*, 2012; Krimpenfort *et al*, 2012).

Here, we report that, in addition to intestinal malignancies, DCC-D1290N mutant mice are prone to develop B-cell type lymphoid

1  Dependence Receptors, Cancer and Development Laboratory - Equipe labellisée 'La Ligue', LabEx DEVweCAN, Centre de Cancérologie de Lyon, INSERM U1052-CNRS UMR5286, Université de Lyon, Centre Léon Bérard, Lyon, France
2  Ecole Nationale Vétérinaire de Lyon, Lyon, France
3  TGF-beta and immune evasion - Centre de Cancérologie de Lyon, INSERM U1052-CNRS UMR5286, Centre Léon Bérard, Lyon, France
4  Service Anatomie et Cytologie pathologiques du Centre Léon Bérard, Lyon, France
5  INSERM, UMR U917, Université Rennes 1, EFS Bretagne, Equipe Labellisée Ligue Contre le Cancer, Rennes, France
6  CHU de Rennes, Pôle Biologie, Rennes, France
7  ANIPATH, Université de Lyon, Lyon, France
8  Service d'Anatomopathologie, Université de Lyon, Hospices Civils de Lyon, Lyon, France
9  Adipogen, Epalinges, Switzerland
10 Pathology of lymphoid cells, Université de Lyon, Service d'Hématologie, Lyon, France
    *Corresponding author. Tel: +33 478782870; e-mail: patrick.mehlen@lyon.unicancer.fr
    **Corresponding author. Tel: +33 478785922; e-mail: marie.castets@lyon.unicancer.fr
    †These authors contributed equally to this work

hyperplasia and follicular (FL) to diffuse large B-cell lymphoma (DLBCL) with age. Furthermore, we show that DCC-induced apoptosis is blocked by loss of DCC expression or by up-regulation of netrin-1, respectively, in germinal center (GC) and in activated B-cell (ABC) DLBCL subtypes. Such high netrin-1 expression levels are also observed in mantle cell lymphoma (MCL) tumors. High netrin-1 expression levels have been described in other cancer pathologies as a mechanism that supports tumor survival by blocking netrin-1 receptor-induced apoptosis (Fitamant *et al*, 2008; Delloye-Bourgeois *et al*, 2009a,b; Gibert & Mehlen, 2015). A therapeutic netrin-1 antibody is under development to hopefully enter phase I clinical trial at the beginning of 2016. Here, we provide the animal proof of concept that netrin-1 interference is sufficient to trigger both ABC-DLBCL and MCL tumor cell death *in vitro* and to induce tumor growth inhibition *in vivo*, thus suggesting that netrin-1 targeting represents an attractive therapeutic approach in these two aggressive subtypes of non-Hodgkin lymphoma.

## Results and Discussion

### Mice lacking DCC-induced apoptosis are susceptible to develop lymphoid hyperplasia and B-cell type lymphoma

As previously established (Mehlen *et al*, 1998; Castets *et al*, 2012), DCC is cleaved by proteases in its intracellular domain (D1290) and this cleavage is a prerequisite for DCC-induced apoptosis (Fig 1A). We have generated a DCC-D1290N knock-in mutant mice model, which presents a specific loss of DCC pro-apoptotic function (Castets *et al*, 2012). Whereas netrin-1 and DCC expression levels are similar in mutant and control animals (Fig 1B), we observed that 44.8% of mutant animals ($n = 29$) display a whole spectrum of lymphoid proliferation abnormalities, ranging from lymphoid hyperplasia to FL and DLBCL. Such lesions were only observed in 15.8% of control mice at 19 months ($n = 19$; Fig 1C and D, and Appendix Fig S1A and B), consistently with previous reports (Frith *et al*, 1993; Festing, 1996). Lesions occurred most prominently in the spleen and lymph nodes, especially the mesenteric ones, but infiltrates were occasionally found in other tissues including liver and kidney (Fig 1C). All hyperplastic and neoplastic cells had a B-cell phenotype (Fig 1E and Appendix Fig S1C), except in one case, which was excluded from incidence calculation. Clonality of B-cell lymphoma was established by the presence of clonal Ig heavy-chain rearrangements (Appendix Fig S1B). Hyperplasias were characterized by extension of follicles (not shown) without disruption of the whole organ architecture, whereas in lymphoma, organ architecture was wiped off by sheets of neoplastic cells (Fig 1E and Appendix Fig S1C). Nodal hyperplasias and DLBCL had a high level of Ki-67 expression, as compared to FL (Fig 1E). Together, these mice data demonstrate that the inhibition of DCC-induced apoptosis is associated with lymphoma development.

### Imbalances in netrin-1/DCC expression ratio are observed in human DLBCL and MCL

We thus investigated whether DCC and netrin-1 levels were modulated in the human pathology. We speculate that silencing of DCC-induced cell death could be associated with human DLBCL occurrence. Classification of DLBCL according to the cell of origin enables those derived from the germinal center (GC) to be distinguished from those derived from activated B-cell type (ABC) (Lenz & Staudt, 2010). We observed that DCC mRNA expression levels are significantly reduced in GC-DLBCL, with a mean expression level of $16.19 \pm 13.37$ in non-tumoral control tonsils ($n = 9$) as compared to $2.77 \pm 3.22$ in GC-DLBCL (5.8-fold decrease; $P = 0.005$, *U*-test; $n = 13$; Fig 2A). On the contrary, DCC mRNA expression remains unchanged in ABC-DLBCL ($n = 18$), but netrin-1 expression is significantly increased by 4.2-fold in this lymphoma subtype ($5.20 \pm 2.79$) compared to normal tonsils ($1.25 \pm 0.40$; $P = 0.004$, *U*-test; Fig 2B). Moreover, all DLBCL samples ($n = 21$) analyzed in a tissue macroarray were positive for netrin-1 expression at protein level, and 71.4% display intermediate to high expression level of this protein ($n = 15/21$; Fig 2C).

Netrin-1 and DCC expression levels were then evaluated in a cohort of other non-Hodgkin lymphoma (NHL) subtypes (data not shown). Since alterations seem to occur in MCL, we screened expression of netrin-1 and DCC mRNA in an independent cohort of MCL biopsies: we confirmed that DCC expression level is indeed significantly reduced by 9.5-fold in MCL (mean expression level of $1.75 \pm 1.32$; $n = 36$) as compared to non-tumoral tonsil samples ($16.19 \pm 13.37$) ($P = 8.5\text{E-}5$, *U*-test; Fig 2A). Reciprocally, netrin-1 mRNA expression is significantly increased by 14.0-fold in MCL ($17.41 \pm 4.15$; $P = 3.11\text{E-}7$, *U*-test; Fig 2B). Such high netrin-1 expression level was confirmed in 55.6% of MCL biopsies ($n = 9$) by immunohistochemistry (Fig 2D).

Altogether, these results indicate that deregulations of DCC and/or netrin-1 expression levels—which globally tip the scales in favor of the inhibition of DCC-induced apoptosis—are associated with DLBCL and MCL occurrence.

### Netrin-1 interference inhibits tumor growth in DLBCL and MCL preclinical models

A large fraction of DLBCL and MCL exhibits gain of netrin-1 expression, which may provide a survival advantage to the tumor by blocking DCC-induced apoptosis. We then first analyzed whether netrin-1 may behave as a survival factor for lymphoma cell lines. We first screened netrin-1/DCC expression in a panel of lymphoma cell lines. ABC-DLBCL cell lines (OCI-Ly3, OCI-Ly10, TMD8, and SUDHL2) express netrin-1 and DCC, whereas expression of this receptor is lower or almost null in the six GC-DLBCL cell lines tested (Fig 3A and Appendix Fig S2A). Netrin-1 was shown to be a transcriptional target of NF-κB (Paradisi *et al*, 2008), which is constitutively active in ABC-DLBCL: we indeed observed that netrin-1 expression is decreased in OCI-Ly3 cells treated with the NF-κB inhibitor Bay 11-7082 (data not shown). Two out of three MCL cell lines were also positive for both ligand and receptor expression. Silencing of netrin-1 expression by siRNA leads to apoptosis engagement in Granta-519 MCL cells as measured by activation of caspase-3, a central effector of apoptosis (Fig 3B and Appendix Fig S2B). Moreover, co-silencing of DCC expression blocks this apoptosis induction, supporting the view that netrin-1 protects these cells from DCC-induced apoptosis (Fig 3B and Appendix Fig S2C) (Fitamant *et al*, 2008; Delloye-Bourgeois *et al*, 2009a,b; Papanastasiou *et al*, 2011). Reciprocally, re-expression of DCC in SUDHL4 negative cell line decreases cell density (data not shown).

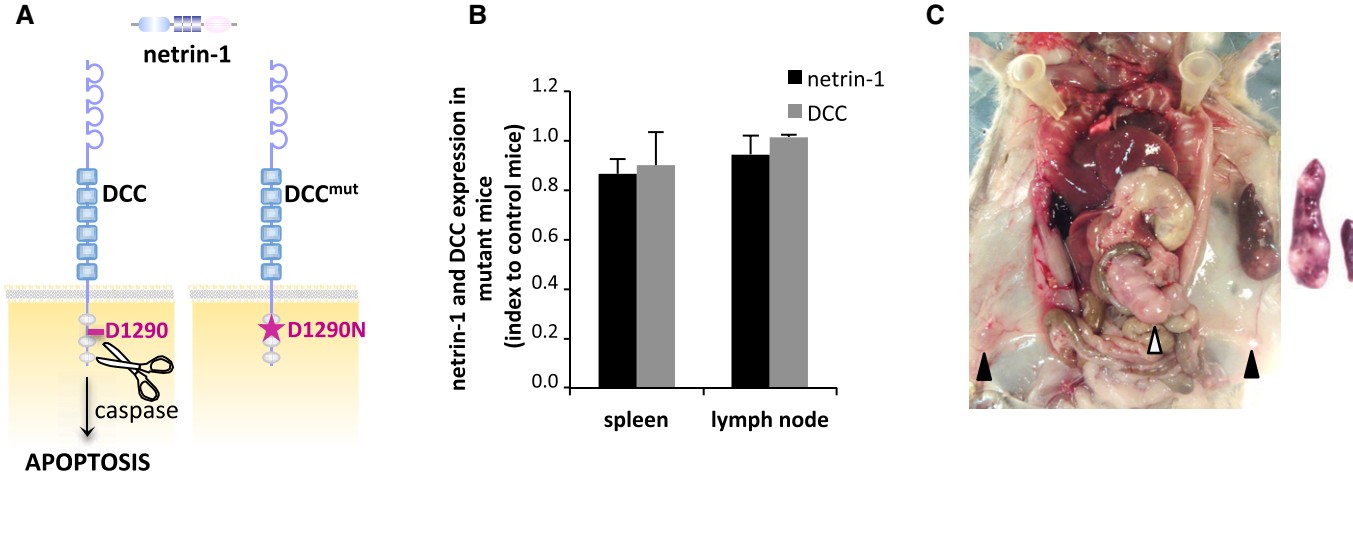

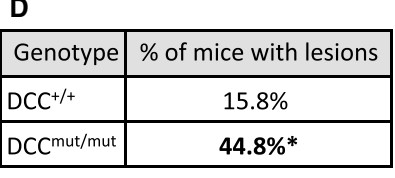

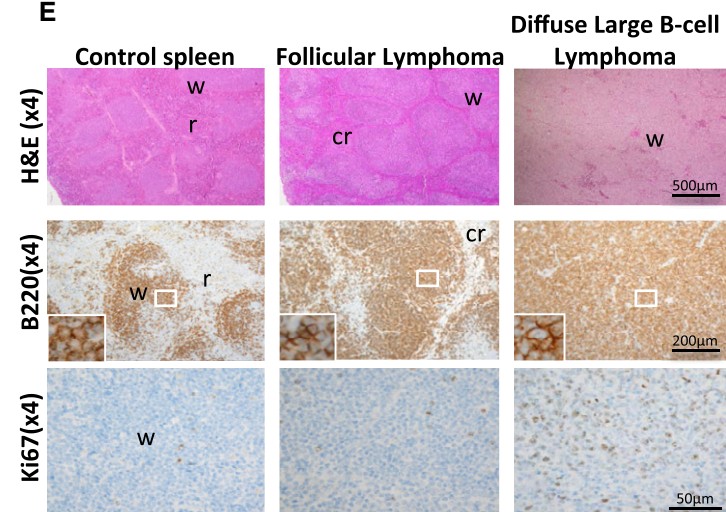

**D**

| Genotype | % of mice with lesions |
|---|---|
| DCC$^{+/+}$ | 15.8% |
| DCC$^{mut/mut}$ | **44.8%*** |

**Figure 1.    Inactivation of DCC-induced apoptosis favors lymphomagenesis in mice.**

A    In the absence of netrin-1, DCC induces apoptosis, unless mutated on its D1290 caspase cleavage site.

B    DCC and netrin-1 expression (FACS analysis) in B-cell lymphocytes isolated from DCC$^{+/+}$ and DCC$^{mut/mut}$ mice spleen and lymph nodes. Results are presented as expression ratios between mean expression levels in mutant animals (*n* = 3) and control ones (*n* = 3). No significant difference was observed between wild-type and mutant animals.

C–E    Incidence and characterization of lymphoid tumoral proliferations in DCC mutant mice. (C) DCC$^{mut/mut}$ mouse with splenomegaly and systemic lymphadenopathy. Left panel: black and white arrows indicate, respectively, the inguinal and mesenteric lymph nodes. Right panel: enlarged spleen in a lymphoma-bearing mutant mouse (left) as compared to control organ (right). (D) Incidence of lymphoid proliferations and lymphoma in DCC$^{+/+}$ (*n* = 19) and DCC$^{mut/mut}$ (*n* = 29) mice. Tumors classification was performed blinded to genotype according to anatomopathologists consensus (Morse *et al*, 2002). *P = 0.036, one-tailed Fisher's exact test. (E) Histological analysis and immunophenotyping of lymphoid proliferations in DCC control and mutant mice. Sections from control spleen, low-grade FL, and high-grade DLBCL were stained with hematoxylin–eosin–safran or with antibodies to B220 or Ki67. w: white pulp, r: red pulp, cr: compressed red pulp.

To move to therapeutic perspective, we analyzed whether a fully human netrin-1 interfering antibody, net-1 mAb, could effectively kill lymphoma cell lines *in vitro* (Fig 3C). Treatment of OCI-Ly3/10 ABC-DLBCL and Granta-519 MCL cell lines with net-1 mAb leads to a decrease in cell density and to an increase in DNA fragmentation, a hallmark of apoptosis observed using a TUNEL assay (Fig 3D and E, and Appendix Fig S2D and E). On the contrary, this compound has no impact on SUDHL4 netrin-1-negative cells, which implies specificity of its cytotoxic effect (Appendix Fig S2F). Incubation of cells with this antibody also induces a significant increase in caspase-3 activity in Granta-519 and OCI-Ly3 cells (Fig 3F and

Appendix Fig S2G). This effect is reversed both by silencing of DCC expression (Fig 3F) and by addition of recombinant netrin-1 (Appendix Fig S2G), confirming the fact that the pro-apoptotic effect of net-1 mAb results from its ability to neutralize netrin-1 and to trigger DCC-induced apoptosis.

We then analyzed whether this can be translated as an anti-tumor effect *in vivo*. We designed murine xenograft models for OCI-Ly3 and Granta-519 cells and observed that intraperitoneal injection of net-1 mAb every other day significantly slows down tumor growth (Fig 3G and H) and increases survival of engrafted animals (Appendix Fig S2H). Moreover, this reduction of tumor growth rate

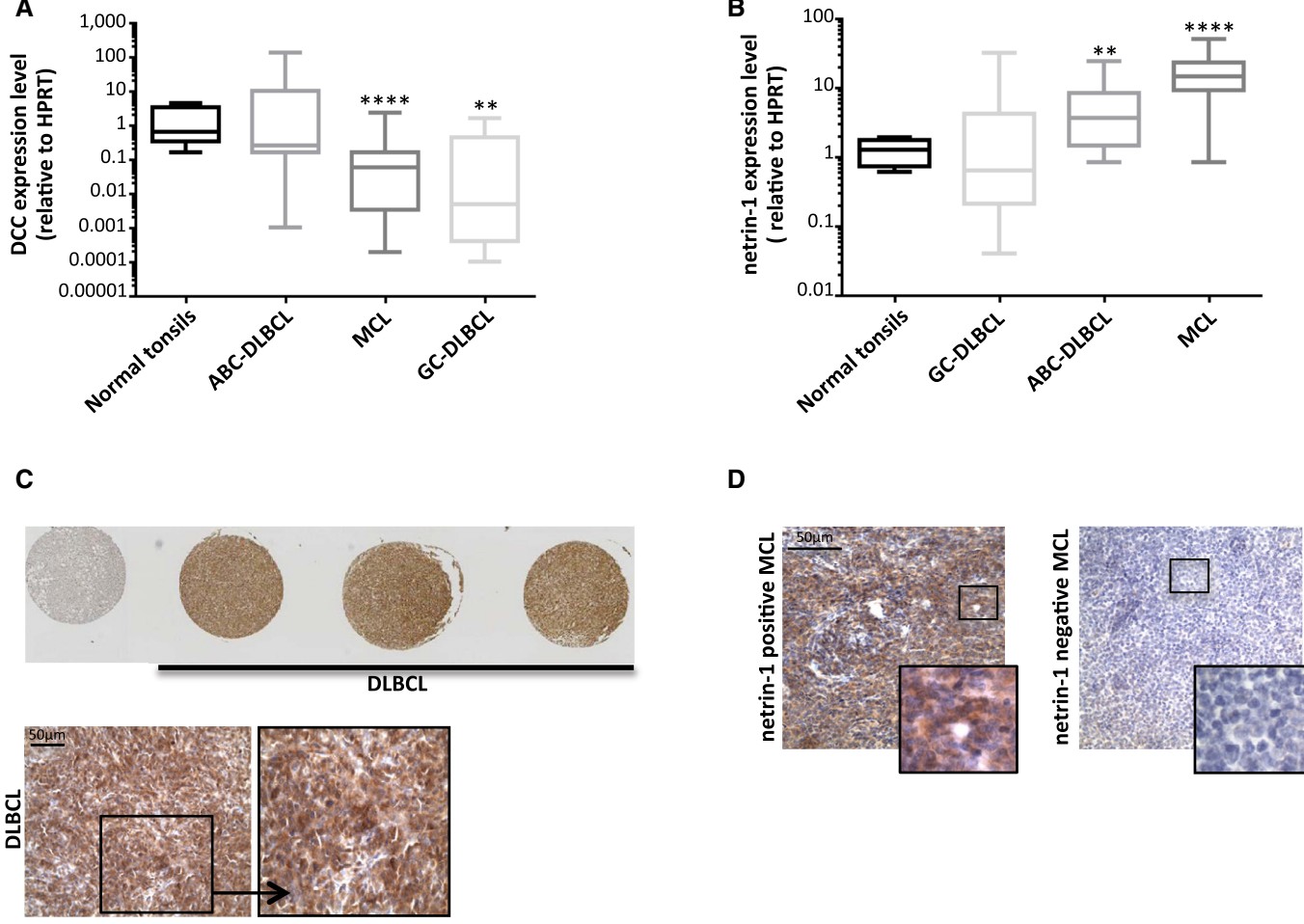

**Figure 2.** **Alterations of netrin-1 and DCC expression levels in human DLBCL and MCL tip the scales toward loss of DCC-induced apoptosis.**

A, B   Analysis of DCC (A) and netrin-1 (B) expressions in a cohort of normal tonsils (*n* = 9), germinal center B-cell-like DLBCL (*n* = 13), activated B-cell-like DLBCL
(*n* = 18), and MCL (*n* = 36). Quantification of genes expression was performed by qRT–PCR, relatively to *HPRT* housekeeping gene. Boxes and whiskers represent the
25–75[th] and 10–90[th] percentiles. **P < 0.005; ****P < 0.0001; two-sided Mann–Whitney *U*-test.
C, D   Analysis of netrin-1 expression in DLBCL (C) and MCL (D) by immunohistochemistry. (C) Twenty-one DLBCL patient biopsies were used to build a TMA. Normal
tissues (normal tonsils, reactive lymph nodes, muscle, liver) are included (the leftmost spot, upper panel, corresponds to liver). Enlarged representative image of
netrin-1 expression in DLBCL is shown (lower panel). (D) Representative images of netrin-1 expression in paraffin-embedded biopsies of MCL (*n* = 9). Five tumors
were positive for netrin-1 (left panel) and four were negative (right panel).

is associated with an increase in caspase-3 activity in tumors
(Fig 3I). Taken together, these data support the view that interfering
with netrin-1/DCC interaction in DLBCL or MCL lymphoma with
netrin-1 expression is associated with tumor cell death and tumor
growth inhibition.

While DCC has been long an enigma in the field of cancer,
being first considered as one of the most important tumor
suppressors and then nearly declassified (Fearon, 1996; Fazeli
*et al*, 1997), several studies have renewed the interest in DCC in
controlling tumor progression. DCC was recently identified as the
third most frequent gene mutated in sun-exposed melanoma
(Krauthammer *et al*, 2012), and its tumor suppressor activity was
established using two different mice models (Castets *et al*, 2012;
Krimpenfort *et al*, 2012). Here, we provide the first demonstration
that specific inhibition of DCC-induced apoptosis in mice is suffi-
cient to induce lymphoid hyperproliferations, ranging from

lymphoid hyperplasia, low-grade FL to high-grade DLBCL, as
already reported for other mice models showing a loss of apop-
totic function, such as Bcl-2-immunoglobulin mutant mice
(McDonnell *et al*, 1989).

Diffuse large B-cell lymphoma is the most common type of
non-Hodgkin lymphoma with an incidence of 7–8 cases/100,000
(Morton *et al*, 2006; Smith *et al*, 2011). Two main entities have
been distinguished among DLBCL: GC, which derives from
normal germinal center B cells, and ABC, which is thought to
result from tumoral transformation of postgerminal center B cells
blocked during plasmacytic differentiation (Leonardo *et al*, 1997).
However, it is now well established that DLBCL is not one but
many different diseases. The same features are reported for MCL,
which is a rare disease accounting for 6–8% of all lymphoma
and is characterized in most of patients by a t(11;14)(q13;q32)
translocation leading to cyclin D1 overexpression (Campo & Rule,

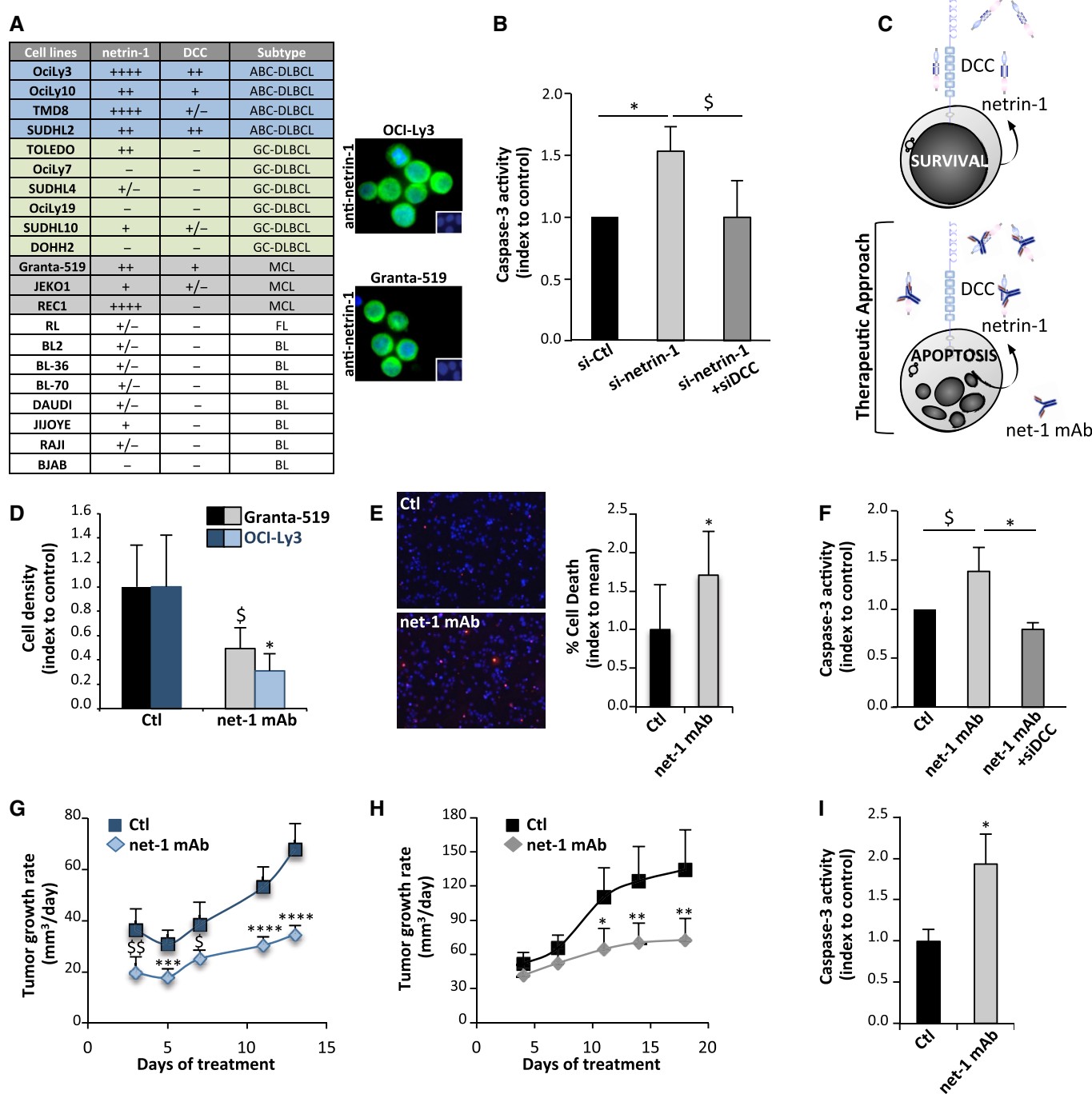

Figure 3.

2015). Even if DLBCL and MCL are two distinct and heterogeneous types of lymphoid neoplasms, we report here a shift toward DCC-induced apoptosis suppression in a fraction of patients suffering from one or the other of these NHL, either by loss of DCC expression, by gain of netrin-1, or by an intermediate system in which both events occur. Although it remains to be understood in a more integrated scheme of pathogenetic mechanisms, we also establish that gain of netrin-1 is a pro-oncogenic event in both ABC-DLBCL and MCL, since netrin-1 behaves as a

survival factor for tumor cells. Despite the use of R-CHOP, those patients with ABC-DLBCL still exhibit inferior outcome and a significant proportion does not respond to this first line treatment. Along the same line and even if the use of rituximab has notably improved the relapse-free and the overall survival of patients, MCL remains in some cases an aggressive disease with poor prognosis. Far from the "one-size-fits-all" vision, the notion that personalized treatment could be an answer to face the heterogeneity of lymphoma has emerged. Although the mechanisms

◀

**Figure 3.   Disruption of netrin-1 autocrine loop with an interfering antibody triggers lymphoma cell death *in vitro* and slows down tumor growth *in vivo*.**

A    Expression of netrin-1 and DCC in lymphoma cell lines. Quantification was performed by qRT–PCR in 21 lymphoma cell lines. *HPRT* housekeeping gene was used as a standardization control. Netrin-1 and DCC levels are indicated as follows: −, not detectable; + to ++++, moderate to very high expression. ABC and GC-DLBCL cell lines are highlighted, respectively, in blue and green, MCL ones are in gray. Expression of netrin-1 (in green) by immunofluorescence using netrin-1 antibody on OCI-Ly3 and Granta-519 cell lines is shown on the right panel. Nuclei are counterstained in blue by Hoechst.

B    Caspase-3 activity in Granta-519 cells after transfection of scrambled siRNA (si-Ctl) or netrin-1 siRNA (si-netrin-1), with or without DCC siRNA (si-DCC). Results are means $\pm$ SD of four independent experiments. $^{\$}P$ = 0.03; $^{*}P$ = 0.02; two-sided Mann–Whitney *U*-test.

C    Schematic model of impact of restoration of DCC-induced cell death by using a netrin-1 targeting antibody.

D    Effect of net-1 mAb on Granta-519 (left panel) and OCI-Ly3 (right panel) cell density. Results are means $\pm$ SD indexed to control of four independent experiments. $^{\$}P$ = 0.05; $^{*}P$ = 0.03; two-sided Mann–Whitney *U*-test.

E    Effect of net-1 mAb on induction of Granta-519 cell apoptosis, detected by TUNEL staining. Left panel: representative images are shown. TUNEL-positive cells are labeled in red. Nuclei are counterstained in blue by Hoechst staining. Right panel: quantification of one representative experiment out of three performed is exposed. Results are presented as percentage of TUNEL-positive cells per field. $^{*}P$ = 0.04; two-sided Mann–Whitney *U*-test.

F    Caspase-3 activity in Granta-519 cells treated with net-1 mAb antibody or with an unrelated Ig-G1 antibody (Ctl), with or without silencing of DCC via siRNA (siDCC). Results are means $\pm$ SD of four independent experiments. $^{\$}P$ = 0.05; $^{*}P$ = 0.02; two-sided Mann–Whitney *U*-test.

G, H    Effect of netrin-1-interfering antibody net-1 mAb on OCI-Ly3 (G) and Granta-519 (H) xenograft tumor growth. Results are presented as means $\pm$ 95% CIs of tumor growth rates from the beginning of treatment. $^{*}P$ = 0.02; $^{**}P$ = 0.01; $^{\$}P$ = 0.007; $^{\$\$}P$ = 0.002; $^{***}P$ = 0.0002; $^{****}P$ < 0.0001; two-sided Student's *t*-test (Granta-519) or Welch *t*-test (OCI-Ly3).

I    Caspase-3 activity in tumors from control ($n$ = 4) or netrin-1-interfering antibody ($n$ = 4)- treated mice. Tumor resection was performed at the end of the treatment. Results are means $\pm$ SD of caspase-3 activity. $^{*}P$ = 0.02; two-sided Mann–Whitney *U*-test.

leading to gain of netrin-1 expression remain to be fully elucidated, the use of a netrin-1 targeting antibody in combination with a chemotherapy backbone could represent an appealing therapeutic strategy in a sizeable fraction of patients suffering from DLBCL or MCL. Along this line, a therapeutic netrin-1 antibody is currently under development. Preliminary preclinical data support the view that this humanized netrin-1 antibody does not show adverse effect in rodents or cynomolgus monkeys and this antibody will enter a phase I clinical trial by early 2016. Thus, the data presented here support the view that DLBCL and MCL could be possible cancer indications for clinical testing of this netrin-1 blocking antibody.

# Materials and Methods

### Patients and tissue samples

A first cohort—comprising 9 normal tonsils, 31 DLBCL (respectively, 18 of activated B-cell and 13 of germinal center B-cell subtypes) and 36 MCL biopsies—was collected by the hematologic division of Rennes University (Hôpital Pontchaillou, Rennes). Samples were frozen (−80°C) immediately after surgery and stored for scientific research in a biological resources repository, according to national ethical guidelines. A second independent cohort of 21 DLBCL biopsies was obtained from the Centre Léon Bérard Hospital (Lyon) and used to build a tissue macroarray (see immunohistochemistry section below). Nine MCL biopsies from the same biological resources center were also used for immunohistochemistry. Tissues banking and researches conducted were approved by the French Ministry of Research. Anatomopathologic characterization of NHL was made according to standard international recommendations.

### Quantitative RT-PCR

Total mRNAs were extracted from frozen tissues/cells using Nucleospin RNAII kit (Macherey-Nagel), and 1 μg was reverse-transcribed using the iScript cDNA Synthesis kit (BioRad).

Real-time quantitative RT–PCR was performed on a LightCycler 2.0 apparatus (Roche) using the Light Cycler FastStart DNA Master SYBERGreen I kit (Roche). Oligonucleotides sequences are available on request.

To assess impact of NF-κB inhibition on netrin-1 expression, OciLy3 cells were seeded into 6-well plates at $3 \times 10^5$ cells per well in a total volume of 2 ml of normal growth medium (see below). NFkB inhibitor (BAY 11-7082) was diluted at a final concentration of 2 μM. Cells were harvested for RNA extraction after 6 hours of incubation with the inhibitor.

### Immunohistochemistry analysis

The anatomopathologic division of Centre Léon Bérard used paraffin-embedded biopsies of a second cohort of 21 DLBCL patients biopsies to build a tissue macroarray (TMA). Normal tissues (normal tonsils, reactive lymph nodes, muscle, liver) were included in the TMA block as controls. Each donor tissue block (21) was punched 3 times for the construction of TMA, which contained 100 tissue cores each. In parallel, 9 paraffin-embedded biopsies of patients with MCL were analyzed. In both cases (DLBCL-TMA and MCL slides), 4-μm-thick sections were sliced; after deparaffinization and rehydration, endogenous peroxidases were blocked by incubating the slides in 5% hydrogen peroxide in sterile water. Heat-induced treatment was performed with 10 mM high pH buffer (Dako) in a PTlink for 30 min. Slides were then incubated in the autostainer (Dako) for 60 min at room temperature with netrin-1 antibody (AF1109, R&D Systems), diluted at 1/100 in antibody diluent solution (Chemmate, Dako). After 3 PBS 1X rinses, slides were incubated with a rabbit anti-goat antibody (E0466, Dako) at 1/200 for 20 min and then with the Flex kit (K800021-2, Dako). Revelation was performed using 3,3-diamino-benzidine, and sections were counterstained with hematoxylin.

Blinded to clinical data, slides were analyzed by the same pathologist, who gauged both the percentage and the intensity (1 + = low, 2 + = intermediate, 3 + = high) of netrin-1 cytoplasmic staining in tumor cells, to determine the percentage of tumors with significant netrin-1 expression at protein level.

## siRNA and reagents

Netrin-1 and DCC siRNAs were designed as a pool of three target-specific siRNAs of 20–25 nucleotides (Santa Cruz biotechnologies, Inc.). Scrambled siRNA that does not trigger degradation of any specific messenger was used as a control (Santa Cruz biotechnologies, Inc.). Human FLAG-tagged netrin-1 was purchased from Adipogen (AG-40B-0040). The netrin-1 mAb (NP114) was provided by Netris Pharma.

## Cell transfection procedure

Granta-519 and SUDHL4 cell lines (from DSMZ) were, respectively, maintained in DMEM or RPMI-Glutamax medium (Invitrogen) supplemented with 10% fetal bovine serum. For siRNA experiments, Granta-519 cells were transfected using Neon transfection system (Neon kit, 10 μl; Invitrogen) using the following parameters: $0.5 \times 10^6$/ml were transfected with 0.8 μg siRNA (0.4 μg of each siRNA) by application of three 1,600 V pulses of 10 ms. SUDHL4 for transfected using the Amaxa system (Lonza) as described previously (Akl *et al*, 2013).

## Cell death assays

Granta-519 cell line (from DMSZ) was maintained in DMEM medium (Invitrogen) supplemented with 10% fetal bovine serum. OCI-Ly3 and 10 cell lines (DSMZ) were cultured in Iscove's modified Dulbecco's medium (IMDM, Invitrogen) supplemented with 20% human serum and 50 μmol/l 2-βmercaptoethanol.

For cell death assays, $0.5 \times 10^6$/ml of Granta-519 or $0.1 \times 10^6$/ml of OCI-Ly3/10 cells were starved, respectively, in DMEM or in IMDM 0% SVF and treated with netrin-1-interfering net-1 mAb antibody or an IgG1-type control antibody at 10 μg/ml in each assay. Netrin-1 was used at 150 ng/ml in all experiments. Cell counting was performed between 24 and 96 h with a NucleoCounter NC-3000 (Beckman). Apoptosis was monitored 24–48 h after treatment using the Caspase 3/CPP32 Fluorimetric Assay Kit (Gentaur Biovision). For detection of DNA fragmentation, treated cells were cytospun and TUNEL assay was performed with 300 U/ml TUNEL enzyme and 6 μM biotinylated dUTP (Roche) as previously described (Castets *et al*, 2009). Human FLAG-tagged netrin-1 and net-1 mAb were obtained from Adipogen (AG-40B-0040).

SUDHL4 cell line was cultured in RPMI-Glutamax medium (Invitrogen) supplemented with 10% fetal bovine serum. After transfection, cells were splitted at $1.25 \times 10^6$/ml in serum-free medium. Cell counting was performed at 120 hours as described above.

## Generation and analysis of DCC-D1290N mutant mice

The DCC-D1290N model was described before (Castets *et al*, 2012). For the analysis of spontaneous neoplasia occurrence, DCC$^{+/+}$ and DCC$^{mut/mut}$ mice were killed at 19 months. Spleen, lymph nodes, and organs with gross lesions were consistently removed. Four-micrometer-thick sections of formalin-fixed and paraffin-embedded samples were first stained with hematoxylin–eosin–safran. In parallel, immunophenotyping of lesions was performed using anti-B-220 (dilution 1:50, BD Pharmingen) and anti-Ki-67 (dilution 1:25, Dako) antibodies. Antigen retrieval was heat-induced. B220 and Ki-67

**The paper explained**

**Problem**

*DCC* (*Deleted in Colorectal Carcinoma*) was considered as one of the most important tumor suppressors in the early 1990s, but nearly declassified at the end of the same decade, after demonstration of its role in nervous system establishment. However, DCC has been recently reinstated as a receptor that constrains tumor progression in breast and colorectal cancers, by inducing apoptosis unless engaged by its ligand netrin-1. DCC was shown to be lost in several cancers among which lymphoma some years ago, but its suppressive function in these cancers is not established.

**Results**

We report here that inhibition of DCC-induced apoptosis is sufficient to increase lymphoma occurrence in a transgenic mice model. In human, loss of DCC-induced apoptosis is retrieved in 2 types of B-cell lymphoma, diffuse large B-cell lymphoma (DLBCL) and mantle cell lymphoma (MCL), either via a decrease in DCC expression or by up-regulation of netrin-1. Using a netrin-1 interfering antibody, we demonstrate both *in vitro* and *in vivo* that netrin-1 acts as a survival factor for a subset of DLBCL and MCL tumor cells.

**Impact**

These data suggest that interference with the netrin-1/DCC interaction could represent a promising therapeutic strategy in netrin-1-positive DLBCL and MCL, which remain therapeutic challenges.

expressions were revealed by brown DAB staining. Nuclei were counterstained by hematoxylin (blue).

Neoplastic lesions were classified and graded in a blinded-fashion according to standard procedures (Morse *et al*, 2002). All experiments were performed in accordance with relevant guidelines and regulations of animal ethics committee (Authorization no CLB_2010_024; accreditation of laboratory animal care by CECCAP, ENS Lyon-PBES).

## Analysis of the clonality of lymphoma

Genomic DNAs were purified from formaldehyde-fixed tissues with FFPE Tissue Kit (Epicentre). The purified DNAs were used as the templates in PCR using DSF and JH4 primers specific for mouse DJ DNA arrangement (Chang *et al*, 1992). DSF primer: 5′-AGGGAT CCTTGTGAAGGGATCTACTACTGTG-3′; JH4 primer: 5′-AAAGACC TCCAGAGGCCATTCTT ACC-3′. PCRs were carried out using Pfu turbo DNA polymerase (Stratagene, La Jolla, CA, USA) under the conditions: 35 cycles of 95°C 30 s, 61.5°C 30 s, 72°C 1 min and followed by 72°C 10 min for extension.

## Flow cytometry analysis of murine B-cell lymphocytes

Single-cell suspensions were prepared from spleen or draining lymph nodes (dLN: pool of inguinal, axillary, and cervical lymph nodes) by manual disruption using glass slides. Isolated cells were pre-incubated with anti-CD16/32 (BD Biosciences) and stained for 30 min at 4°C with the following antibodies: anti-CD19 (BD Biosciences), netrin-1 antibody (AF1109, R&D Systems), or DCC antibody (A-20, Santa Cruz Biotechnologies, Inc.). Cells were washed and incubated with Alexa Fluor 488 Donkey Anti-Goat IgG (H+L) Antibody (Life Technologies) for a further 30 min at 4°C after

the primary staining. Flow cytometric analysis was performed using a CyAn ADP Analyzer (Beckman Coulter). Data analysis was carried out using FlowJo software (Tree star Inc.). The positive staining gate was set using anti-goat IgG in the absence of the primary antibody.

### Xenografts of Granta-519 and Oci-LY3 in immunodeficient mice

Five-week-old female nude (Granta-519) or SCID (OCI-Ly3) mice were obtained from Charles River laboratories. Mice were housed in sterilized filter-topped cages and maintained in a pathogen-free animal facility. Granta-519 and OCI-Ly3 cells were implanted by subcutaneous injection of, respectively, $3.0 \times 10^6$ cells in 100 μl of PBS or $4 \times 10^6$ cells in 100 μl of growth factors reduced matrigel (Corning) diluted in 100 μl of PBS into the left flank of the mice. When tumors reached 150 mm$^3$ (Granta-519) or became palpable (OCI-Ly3), mice with size-matched tumors were randomly split in two groups blinded to treatment assignment and received each two days intraperitoneal injections of net-1 mAb at 20 mg/kg ($n = 12$ for Granta-519, $n = 30$ for Oci-Ly3), or an equal volume of control IgG1 ($n = 9$ for Granta-519, $n = 30$ for Oci-Ly3). Tumor sizes were measured with a caliper. Tumor volumes were calculated with the formula $v = 0.5 \times (l \times w^2)$, where $v$ is volume, $l$ is length, and $w$ is width. Apparition of necrosis, tumor superior to 17 mm in length or tumor volume exceeding 2,000 mm$^3$ defined experimental endpoints, according to ethical guidelines. All experiments were performed in accordance with relevant guidelines and regulations of animal ethics committee (Authorization no CLB_2014_009; accreditation of laboratory animal care by CECCAP, ENS Lyon-PBES). For caspase-3 activity measurement, xenografted tumors were resected at the end of treatment and enzymatic activity of this cysteine protease was measured on whole protein lysates following manufacturer's instructions (Gentaur, Biovision).

### Statistical methods

Statistical significance of differences between groups was evaluated by *t*-test (Welch correction was used in case of variances inequality under *F*-test), Mann–Whitney test, or Fisher's exact test. *P* values < 0.05 were considered to be statistically significant.

**Expanded view** for this article is available online.

### Acknowledgments
We wish to thank the LMT platform for their support with animal work and Yohann Chaix and Peter Mulligan for critical reading of the manuscript. We are also grateful to the Centre de Ressources Biologiques-Santé (BB-0033-00056, http://www.crbsante-rennes.com) of Rennes hospital for its support in the processing of biological samples. This work was supported by institutional grants from CNRS (PM), University of Lyon (PM), Centre Léon Bérard (PM) and from the Ligue Contre le Cancer (PM, KT), INCA (PM), ANR (PM), ERC (PM), Fondation Bettencourt (PM), and ITMO Cancer (Lymphoma, MC).

### Author contributions
LB, MC, JGD and JV have performed experimental design and work, with the help of GC, MJMC, and JB. AT and NG have realized anatomopathological analyses of murine samples. ATG, CCC, GS, and KT have provided human biopsies and cell lines used in this report and performed analysis of netrin-1/DCC expression level, with the technical support of SL, CP, AC, and MLB. JCM has provided some scientific insights about FACS analysis. PM and MC proposed the project, did experimental design, and wrote the manuscript.

### Conflict of interest
Patrick Mehlen declares to have conflict of interest as shareholder of Netris Pharma.

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
