## [Review Process File · EMBO Molecular Medicine]

Targeting netrin-1/DCC interaction in Diffuse Large-B and Mantle Cell Lymphoma.

Laura Broutier, Marion Creveaux, Jonathan Vial, Antonin Tortereau, Guillaume Chazot, Mark MacCarron, Sophie Léon, Céline Pangault, Nicolas Gadot, Amélie Colombe, Marie-Laure Boulland, Jonathan Blachier, Julien C. Marie, Alexandra Traverse-Glehen, Olivier Donzé, Catherine Chassagne-Clément Gilles Salles, Karin Tarte, Patrick Mehlen and Marie Castets

Corresponding authors: Dr. Patrick Mehlen and Dr. Marie Castets, Université de Lyon, Centre Léon Bérard

Review timeline:	Submission date:	03 June 2015
	Editorial Decision:	06 July 2015
	Revision received:	20 October 2015
	Editorial Decision:	09 November 2015
	Revision received:	24 November 2015
	Accepted:	04 December 2015

Transaction Report:

Editor: Roberto Buccione

1st Editorial Decision

06 July 2015

Thank you for the submission of your manuscript to EMBO Molecular Medicine. We have now heard back from the three Reviewers whom we asked to evaluate your manuscript.

You will see that the Reviewers are mostly supportive of your work although they do express a number of concerns that prevent us from considering publication at this time. I will not dwell into much detail, as the evaluations are self-explanatory. I would like, however, to highlight a few main points.

Reviewer 2 would like you to strengthen your conclusions by including additional cell lines and also to include an important control to confirm the specificity of the cytotoxic effect of the netrin-1 mAb. This Reviewer also suggest that it would be of interest to verify Netrin-1 levels after NF- κ B inhibition to explore the role of the latter. I suggest inclusion of these data, especially if available, but would consider this point to be further-reaching with respect to the general message of the manuscript.

Reviewer 3 is clearly more reserved. S/he asks a number of pertinent questions finalised towards establishing the extent of the effects observed and clarifying important experimental details and rationale. S/he would also like to see the primary data for the DCC/Netrin-1 expression in the cell lines, and as also requested by Reviewer 2, validation of netrin-1 mAb toxicity in additional cell lines.

Considered all the above, while publication of the paper cannot be considered at this stage, we would be pleased to consider a revised submission, with the understanding that the Reviewers' concerns must be addressed as outlined above, with additional experimental data where appropriate and that acceptance of the manuscript will entail a second round of review.

It is EMBO Molecular Medicine policy to allow a single round of revision only and that, therefore, acceptance or rejection of the manuscript will depend on the completeness of your responses included in the next, final version of the manuscript.

As you know, EMBO Molecular Medicine has a "scooping protection" policy, whereby similar findings that are published by others during review or revision are not a criterion for rejection. However, I do ask you to get in touch with us after three months if you have not completed your revision, to update us on the status. Please also contact us as soon as possible if similar work is published elsewhere.

Please note that EMBO Molecular Medicine now requires a complete author checklist (<http://embomolmed.embopress.org/authorguide#editorial3>) to be submitted with all revised manuscripts. Provision of the author checklist is mandatory at revision stage; The checklist is designed to enhance and standardize reporting of key information in research papers and to support reanalysis and repetition of experiments by the community. The list covers key information for figure panels and captions and focuses on statistics, the reporting of reagents, animal models and human subject-derived data, as well as guidance to optimise data accessibility.

Finally, please upload figures as individual files when submitting your revision and make sure you adhere to our new Author guidelines for provision of expanded view figures (formerly supplementary information; <http://embomolmed.embopress.org/authorguide#expandedview>).

I look forward to seeing a revised form of your manuscript in due time.

***** Reviewer's comments *****

Referee #1 (Comments on Novelty/Model System):

The DCC gene has been described in 1990 as a tumor suppressor gene deleted in colon cancer (DCC) by Vogelstein's group, but its role as a tumor suppressor gene has long been disputed. DCC turned out to be a receptor that promotes neuronal guidance and repulsion upon binding of its ligand Netrin-1. The interest in DCC as a tumor suppressor gene has revived with the discovery that DCC serves as a conditional cell death receptor (dependence receptor) that induces apoptosis in different cell types in the absence of its ligand and promotes survival upon binding of Netrin-1. Induction of apoptosis is linked to cleavage of DCC at position D1290. The D1290N mutation abolishes cleavage and by doing so abrogates the pro-apoptotic function of DCC. The authors have created a mouse model with mutated DCC D1290N in which the pro-apoptotic function of DCC is lost while the survival function is uncompromised. Analysis of spontaneous occurrence of neoplasias in these mice revealed that the DCC-D1290N mice displayed a threefold higher incidence of B cell neoplasias as compared to control mice at the age of 19 months with the whole spectrum of tumors (B cell hyperplasia, follicular lymphoma and diffuse large cell B cell lymphoma (DLBCL)). The analysis of DCC and Netrin-1 expression on neoplastic B cells revealed distinct patterns in different tumor types: significantly decreased expression of DCC on mantle cell (MCL) and diffuse large B cell lymphomas of the germinal center type (GC-DLBCL) and no change in DLBCL of the activated B cell (ABC) type. Expression analysis of Netrin-1 revealed a reciprocal pattern: significantly higher expression in ABC-DLBCL and MCL (ABC-DLBCL > MCL), whereas the expression level of Netrin-1 was unchanged in GC-DLBCL. The ratio of Netrin-1 to DCC was thus significantly increased in all B cell lymphoma subtypes observed. Apoptosis was induced by siRNA knockdown of Netrin-1 in those ABC-DLBCL and MCL cell lines that express Netrin-1 and this could be

reverted by concomitant knockdown of DCC. Apoptosis could also be induced by a monoclonal antibody against Netrin-1 and this effect was likewise reversed by knockdown of DCC. The anti-Netrin-1 monoclonal antibody slowed down proliferation of the ABC-DLBCL cell line OCI-Ly3 and the MCL cell line Granta-519 in a xenograft mouse model and prolonged survival to a significant extent.

The finding that the Netrin-1/DCC-systems play also an important role in a significant fraction of human B cell lymphomas makes Netrin-1 an interesting drug target for reactivation of the cell death pathway in these patients. The work is of high quality. The data support the conclusions drawn. I recommend this manuscript for publication.

Referee #2 (Remarks):

Comments to authors:

The manuscript by Broutier et al. addresses the role of netrin-1 and DCC in the development of Diffuse Large-B (DLBCL) and Mantle Cell (MCL) Lymphoma. Using a mouse model that lacks the proapoptotic DCC activity, they found a tumor suppressor activity of the netrin-1/DCC axis in the development of lymphomas. Expression profiling of DCC and netrin-1 in human lymphoma biopsies strengthens their hypothesis for an important role of netrin-1/DCC. Targeting the netrin-1/DCC axis by an siRNA-mediated silencing approach or by neutralizing netrin-1 antibody shows induction of cell death in human DLBCL and MCL cell lines and therefore suggests therapeutic potential.

These findings are very interesting and the data are of high technical quality. Also the combined use of cell lines, mouse models and human biopsies addresses the relevant questions and is state of the art.

Nevertheless, I have a few concerns that should be addressed by the authors.

Major concerns:

- 1) The high incidence of spontaneous B-cell lymphomas in control mice (16%) is very surprising. Please comment and indicate the age of the mice in the results section.
- 2) Maybe I misunderstand but to me the data in Suppl Fig. 1 suggest that both the spleen of normal mice and the lymphoma of mutant mice contained B-cells of clonal origin? Or has the annotation been inverted?
- 3) The authors should include additional ABC-DLBCL cell lines in Figure 3A (such as HBL-1 or TMD8). The use of more than two cell lines of this subtype would strengthen the model.
- 4) To support the specific cytotoxic effect of the netrin-1 mAb, REC1 or SUDHL4 cells, which produce netrin-1 but no DCC, should be incubated with the antibody. This experiment, together with Figure 3B and 3F would exclude toxic side effects of the netrin-1 mAb.
- 5) ABC DLBCL and a subset of mantle cell lymphoma have constitutive activation of the NF-kB pathway. It would be interesting to quantify netrin-1 levels after NF-kB inhibition to address whether NF-kB drives the netrin-1 expression.

Minor points:

- 1) The total number of control and mutant mice analyzed should be indicated in Suppl Fig. 1
- 2) It is not clear which normal tissue (the text mentions normal tonsil, reactive lymph node, muscle,

liver) is included in the upper panel of Fig. 2C.

3) The manuscript contains many small grammatical errors that need to be corrected.

Referee #3 (Comments on Novelty/Model System):

These findings could be of interest. However, unfortunately the current version of the manuscript is poorly written. It is not clear which analyses have been performed and how many samples have been investigated (see below). In addition, various primary data need to be shown as figures as well additional controls have to be performed (see below). Due to these shortcomings it is difficult to review the manuscript in its current version. Thus, I feel that the manuscript by Broutier et al. needs to be revised substantially to be suitable for publication in EMBO Molecular Medicine. Please find a point-by-point criticism below.

Referee #3 (Remarks):

Broutier and colleagues present data on the role of the DCC (Deleted in Colorectal Carcinoma) network in the biology of diffuse large B-cell lymphoma (DLBCL) and mantle cell lymphoma (MCL). The authors suggest that interference with DCC in a mouse model is associated with lymphomagenesis. The authors show that DCC levels are reduced in the germinal center B-cell like subtype of DLBCL (GCB DLBCL) and in MCL, whereas the DCC ligand netrin-1 is overexpressed in the activated B-cell like subtype (ABC DLBCL) and in MCL. Finally, the authors show that a human netrin-1 antibody was toxic to ABC and MCL cell lines and xenograft mouse models.

Major criticism:

1. Was the difference in lymphoma development between the DCC-D1290N knock in mouse model and the control mice (44.8% vs. 15.8%) significant? In addition, the authors need to clearly specify the percentages of lymphoma subtypes that were observed, i.e. how frequent were lymphoid hyperplasias, FL, and DLBCL.
2. The paragraph on the "Imbalance in netrin-1/DCC expression..." on page 6 is unclear. The authors state that DCC levels were significantly reduced in GCBs. What was measured here, mRNA or protein? How many samples were investigated? What were the controls? The same applies for the other expression analyses.
3. What does "intermediate to high expression of netrin-1" as determined by IHC mean? This needs to be defined and explained why this strategy was applied.
4. On page 7 it is stated that expression of DCC/netrin-1 was investigated in a panel of cell lines. What was assessed protein or RNA? How was this determined? The primary data need to be shown.
5. For the siRNA experiments the knockdown data on both RNA and protein need to be shown. More than just one cell line need to be investigated for induced toxicity.
6. To show that DCC is involved in the biology, DCC should be expressed in deficient cell lines to investigate if apoptosis is induced by re-expression of DCC.
7. In the experiments with the netrin-1 interfering antibody additional cell lines and importantly at least one negative control cell line need to be investigated to show the specificity of the approach.

Lyon, October 12th, 2015

The Editor

EMBO MOLECULAR MEDICINE

ms: **EMM-2015-05480**

Dear Editor,

Thank you for your kind comments on our manuscript, “**Targeting netrin-1/DCC interaction in Diffuse Large-B and Mantle Cell Lymphoma**” by Broutier et al. We have endeavored to address the points made by the three referees.

Referee #1:

“The finding that the Netrin-1/DCC-systems play also an important role in a significant fraction of human B cell lymphomas makes Netrin-1 an interesting drug target for reactivation of the cell death pathway in these patients. The work is of high quality. The data support the conclusions drawn. I recommend this manuscript for publication.”

We thank the referee for his/her kind comments.

Referee #2:

“ These findings are very interesting and the data are of high technical quality. Also the combined use of cell lines, mouse models and human biopsies addresses the relevant questions and is state of the art. Nevertheless, I have a few concerns that should be addressed by the authors”.

We thank the referee for his/her kind comments and have addressed the comments as described below

Major concerns:

“1) The high incidence of spontaneous B-cell lymphomas in control mice (16%) is very surprising. Please comment and indicate the age of the mice in the results section.”

Mice were analysed at 19 months of age. This is now more adequately indicated. The incidence observed in control mice is however in accordance with previous reports of spontaneous occurrence of such lesions in the C57Bl6 strain during ageing (The Morphology, Immunohistochemistry and Incidence of Hematopoietic Neoplasms in mice and rats, Frith et al., 1993; Origins and characteristics of inbred strains of mice, Festing M., 1996). This has been indicated in the manuscript.

“2) Maybe I misunderstand but to me the data in Suppl Fig. 1 suggest that both the spleen of normal mice and the lymphoma of mutant mice contained B-cells of clonal origin? Or has the annotation been inverted?”

We agree that it was not appropriately explained: The amplification of three PCR fragments (DJH1, DJH2 and DJH3) indicates that different populations of B-cells coexist in normal mice spleen, which supports the view of multiples clones in the spleen of normal mice. On the contrary, prevalence of one single band in the DCC mutant mice is in favour of the clonal origin of the lesion observed. This is now more clearly explained in Fig S1 legend.

“3) The authors should include additional ABC-DLBCL cell lines in Figure 3A (such as HBL-1 or TMD8). The use of more than two cell lines of this subtype would strengthen the model.”

Additional cell lines, and notably TMD8, have been included in Figure 3A. They all express netrin-1 and DCC.

“4) To support the specific cytotoxic effect of the netrin-1 mAb, REC1 or SUDHL4 cells, which produce netrin-1 but no DCC, should be incubated with the antibody. This experiment, together with Figure 3B and 3F would exclude toxic side effects of the netrin-1 mAb.”

Netrin-1 mAb has been tested on SUDHL4 cells that do not express DCC. No significant impact on their viability has been observed, which indeed support the expected mode of action and strengthens the specificity of antibody (Fig S2F).

“5) ABC DLBCL and a subset of mantle cell lymphoma have constitutive activation of the NF- κ B pathway. It would be interesting to quantify netrin-1 levels after NF- κ B inhibition to address whether NF- κ B drives the netrin-1 expression.”

We agree. Indeed, constitutive activation of the NF- κ B pathway is observed in ABC subgroup of DLBCL tumors and cell lines, such as Oci-Ly3, and netrin-1 was shown to be a transcriptional target of NF- κ B in inflammatory colorectal cancers (Paradisi et al., 2008, Gastroenterology). After 6 hours of treatment with the NF- κ B inhibitor Bay 11-7082 at 2 μ M, we observed that netrin-1 expression is decreased by 1.7 fold in Oci-Ly3 cells (see below), suggesting that netrin-1 activation could at least partially result from NF- κ B activation in some lymphoma cell lines. This information has been added as “*data not shown*” in the Results section.

Effect of NF- κ B inhibitor Bay 11-7082 on netrin-1 expression in Oci-Ly3 ABC-DLBCL cells. Cells were treated 6 hours with or without Bay 11-7082 at 2 μ M. After mRNA extraction, netrin-1 and I- κ B expression levels were analysed by Q-RT-PCR using specific primers. I- κ B expression is used as a positive control, since it is a direct transcriptional target of NF- κ B. Results are mean \pm std.

Minor points:

“1) The total number of control and mutant mice analyzed should be indicated in Suppl Fig. 1”

This information has been added.

“2) It is not clear which normal tissue (the text mentions normal tonsil, reactive lymph node, muscle, liver) is included in the upper panel of Fig. 2C.”

Spot of normal tissue in Fig 2C corresponds to liver. This information has been added.

“3) The manuscript contains many small grammatical errors that need to be corrected.”

The manuscript has now been corrected by an english native speaker.

Referee #3:

“These findings could be of interest. However, unfortunately the current version of the manuscript is poorly written. It is not clear which analyses have been performed and how many samples have been investigated (see below). In addition, various primary data need to be shown as figures as well additional controls have to be performed (see below). Due to these shortcomings it is difficult to review the manuscript in its current version. Thus, I feel that the manuscript by Broutier et al. needs to be revised substantially to be suitable for publication in EMBO Molecular Medicine. Please find a point-by-point criticism below.”

We thank the referee for his/her comments and have addressed his/her comments as described below

Major criticism:

“1. Was the difference in lymphoma development between the DCC-D1290N knock in mouse model and the control mice (44.8% vs. 15.8%) significant? In addition, the authors need to clearly specify the percentages of lymphoma subtypes that were observed, i.e. how frequent were lymphoid hyperplasias, FL, and DLBCL.”

The difference between DCC-D1290N and control mice is indeed significant using one-tailed Fisher exact test at 5% level. Respective frequencies have been added in Fig S1A.

“2. The paragraph on the "Imbalance in netrin-1/DCC expression..." on page 6 is unclear. The authors state that DCC levels were significantly reduced in GCBs. What was measured here, mRNA or protein? How many samples were investigated? What were the controls? The same applies for the other expression analyses.”

We agree that the initial manuscript was missing important information. DCC expression was assessed at mRNA level in 13 GCB tumor samples versus 9 non-tumoral tonsils. This is now stated in Figure 2A legend.

“3. What does "intermediate to high expression of netrin-1" as determined by IHC mean? This needs to be defined and explained why this strategy was applied.”

Netrin-1 expression level has been gauged in blind by experienced pathologists in lymphoma and scored as low, intermediate or high as compared to that in control samples, in order to strengthen data obtained by Q-RT-PCR. This is the usual +/-, +, ++ indication used by pathologists for clinical annotation that has also previously used for netrin-1 staining (Delloye et al., JNCI 2009). This is now more adequately described in Material and Methods sections.

“4. On page 7 it is stated that expression of DCC/netrin-1 was investigated in a panel of cell lines.

What was assessed protein or RNA? How was this determined? The primary data need to be shown.”

We apologize for the lack of information in the initial manuscript. Netrin-1 and DCC expressions were quantified at mRNA level using Q-RT-PCR. Primary quantification has been added in Fig S2A. Expression at the endogenous protein level has unfortunately always been an issue for researcher working on netrin-1/receptors because of (i) the relative low expression of these proteins and (ii) the absence of adequate tools. However even though the expression has been mainly reported based on RNA, the function of the product of these RNA has been extensively described by us and others.

“5. For the siRNA experiments the knockdown data on both RNA and protein need to be shown. More than just one cell line need to be investigated for induced toxicity.”

Detection of endogenous DCC/netrin-1 by western-blot has already been reported as technically very tricky, thereby preventing evaluation of the efficiency of the siRNA approach used at protein level (see also point 4). However, we have shown at mRNA level that the siRNA strategy efficiently silences the expression of netrin-1 and DCC (Fig S1B-C) and that this silencing is associated with functional effects.

“6. To show that DCC is involved in the biology, DCC should be expressed in deficient cell lines to investigate if apoptosis is induced by re-expression of DCC.”

We have now shown that transient transfection of DCC in SUDHL4 cell line, which expresses low level of this receptor, has indeed a direct biological effect on cell death (see below, inserted as “*data not shown*” in the manuscript). Re-expression of DCC inducing tumor cell death has been shown extensively in the past (Mehlen et al., Nature 1998; Forcet et al., PNAS, 2001; Castets et al., Nature, 2012)

Effect of DCC reexpression on SUDHL4 cell density. Transfection was performed using AMAXA nucleofection system as previously described (Akl et al., Cell Death & Disease, 2013). Results are mean +/- std.

“7. In the experiments with the netrin-1 interfering antibody additional cell lines and importantly at least one negative control cell line need to be investigated to show the specificity of the approach.”

This is an important point indeed. In addition to the three positive cell lines that were shown to be sensitive to netrin-1 mAb (GRANTA-519, Oci-Ly3 and Oci-Ly10), netrin-1 mAb has been tested on SUDHL4 cells that do not express DCC. No significant impact of netrin-1 mAb on their viability has been observed, which strengthens the specificity of this compound (Fig S2F).

We are grateful to the reviewers for their comments, which we believe have strengthened the manuscript. We believe that this manuscript provides new insights in the fields of oncology and apoptosis.

If we should send additional information, please let me know. We thank you in advance for your consideration of the revised manuscript.

Thank you for the submission of your revised manuscript to EMBO Molecular Medicine. We have now received the enclosed reports from the Reviewers that were asked to re-assess it. As you will see the reviewers are now supportive and I am pleased to inform you that we will be able to accept your manuscript pending the following final amendments:

- 1) Please note and take action on Reviewer 1's comment concerning Fig. 1A.
- 2) As per our Author Guidelines, the description of all reported data that includes statistical testing must state the name of the statistical test used to generate error bars and P values, the number (n) of independent experiments underlying each data point (not replicate measures of one sample), and the actual P value for each test (not merely 'significant' or 'P < 0.05').
- 3) Every published paper now includes a 'Synopsis' to further enhance discoverability. Synopses are displayed on the journal webpage and are freely accessible to all readers. They include a short standfirst as well as 2-5 one sentence bullet points that summarise the paper. Please provide the synopsis including the short list of bullet points that summarise the key NEW findings. The bullet points should be designed to be complementary to the abstract - i.e. not repeat the same text. We encourage inclusion of key acronyms and quantitative information. Please use the passive voice. Please attach this information in a separate file or send them by email, we will incorporate it accordingly. You are also welcome to suggest a striking image or visual abstract to illustrate your article. If you do please provide a jpeg file 550 px-wide x 400-px high.
- 4) The Supporting information must be changed to our new format (<http://embomolmed.embopress.org/authorguide>). In brief: Extra figures..... and extra text (e.g. extra methods) should be provided in a single PDF (nomenclature to name and refer to Appendix items in the main text: Appendix Figure S1, Appendix Table S1, Appendix Supplementary Methods). The Appendix should begin with a short table of contents. The Appendix will be provided at the end of the manuscript as PDF. UPLOAD as a single PDF using the file type Expanded View File

I look forward to reading a new revised version of your manuscript as soon as possible.

***** Reviewer's comments *****

Referee #1 (Comments on Novelty/Model System):

I have outlined my opinion on the manuscript in my previous review. Netrin-1 turns out to be a very interesting novel drug target for human high grade B cell lymphomas.
The revision has still further strengthened the manuscript.
I recommend the manuscript for publication.

One minor point:

in the left part of Figure 1A: shouldn't it read D1290 instead of D1290N?

Referee #2 (Remarks):

The authors provide additional data, which strengthen their claims. All open questions have been answered so that I can recommend the current manuscript for publication.

Referee #3 (Remarks):

The authors have adequately addressed the raised concerns.